# Characterization of IL-2 Stimulation and TRPM7 Pharmacomodulation in NK Cell Cytotoxicity and Channel Co-Localization with PIP_2_ in Myalgic Encephalomyelitis/Chronic Fatigue Syndrome Patients

**DOI:** 10.3390/ijerph182211879

**Published:** 2021-11-12

**Authors:** Stanley Du Preez, Natalie Eaton-Fitch, Helene Cabanas, Donald Staines, Sonya Marshall-Gradisnik

**Affiliations:** 1National Centre for Neuroimmunology and Emerging Diseases, Menzies Health Institute, Griffith University, Gold Coast 4215, Australia; natalie.eaton-fitch@griffithuni.edu.au (N.E.-F.); d.staines@griffith.edu.au (D.S.); s.marshall-gradisnik@griffith.edu.au (S.M.-G.); 2Consortium Health International for Myalgic Encephalomyelitis, Menzies Health Institute Queensland, Griffith University, Gold Coast 4215, Australia; helene.cabanas@inserm.fr; 3School of Pharmacy and Medical Sciences, Griffith University, Gold Coast 4215, Australia; 4School of Medicine and Dentistry, Griffith University, Gold Coast 4215, Australia; 5INSERM U944 and CNRS UMR 7212, Institut de Recherche Saint Louis, Hôpital Saint Louis, APHP, Université de Paris, 75010 Paris, France

**Keywords:** myalgic encephalomyelitis, chronic fatigue syndrome, natural killer cell, transient receptor potential melastatin 7, IL-2, PIP_2_

## Abstract

Myalgic encephalomyelitis/chronic fatigue syndrome (ME/CFS) is a complex multisystemic disorder responsible for significant disability. Although a unifying etiology for ME/CFS is uncertain, impaired natural killer (NK) cell cytotoxicity represents a consistent and measurable feature of this disorder. Research utilizing patient-derived NK cells has implicated dysregulated calcium (Ca^2+^) signaling, dysfunction of the phosphatidylinositol-4,5-bisphosphate (PIP_2_)-dependent cation channel, transient receptor potential melastatin (TRPM) 3, as well as altered surface expression patterns of TRPM3 and TRPM2 in the pathophysiology of ME/CFS. TRPM7 is a related channel that is modulated by PIP_2_ and participates in Ca^2+^ signaling. Though TRPM7 is expressed on NK cells, the role of TRPM7 with IL-2 and intracellular signaling mechanisms in the NK cells of ME/CFS patients is unknown. This study examined the effect of IL-2 stimulation and TRPM7 pharmacomodulation on NK cell cytotoxicity using flow cytometric assays as well as co-localization of TRPM7 with PIP_2_ and cortical actin using confocal microscopy in 17 ME/CFS patients and 17 age- and sex-matched healthy controls. The outcomes of this investigation are preliminary and indicate that crosstalk between IL-2 and TRMP7 exists. A larger sample size to confirm these findings and characterization of TRPM7 in ME/CFS using other experimental modalities are warranted.

## 1. Introduction

Natural killer (NK) cells are granular lymphocytes that provide anti-viral and anti-tumor defense primarily as part of the innate immune response. The two major human NK cell subsets include cluster of differentiation (CD)56^bright^CD16^dim/-^ and CD56^dim^CD16^+^ NK cells [1]. Immunosurveillance and elimination of transformed autologous cells through cytotoxic effector mechanisms are principally performed by CD56^dim^CD16^+^ NK cells circulating in peripheral blood [2,3,4]. NK cell cytotoxicity is mediated by the degranulation of cytotoxic effector molecules including perforin, granzyme, and death ligands stored within secretory lysosomes [5,6,7,8,9]. Importantly, interleukin-2 (IL-2) stimulation significantly enhances the reactivity of NK cells with target cells via a mechanism termed lymphokine-activated killing (LAK) [10,11,12,13].

IL-2 binds the IL-2 receptor (IL-2R) expressed by NK cells and a variety of other immune cells. CD56^dim^CD16^+^ NK cells specifically express the intermediate affinity IL-2Rβ, which associates with the common gamma chain subunit forming a heterodimeric signalling complex [10,14]. Downstream mediators of IL-2R activation enhance the transcription of proteins, including cytotoxic effector molecules, that enhance the killing capacity of NK cells [10,13,15]. Once a suitable target cell is identified, an immune synapse (IS) is formed, and the concerted polarization and degranulation of secretory lysosomes within NK cells is regulated by calcium (Ca^2+^) [16].

Ca^2+^ is an essential cellular signaling molecule that supports several processes including cell differentiation, activation, apoptosis, and NK cell degranulation [17]. Degranulation processes are activated by increased intracellular Ca^2+^ following the release of intracellular Ca^2+^ stores and influx through Ca^2+^-permeable membrane ion channels. Ca^2+^ enhances transcriptional events requisite to NK cell cytotoxicity and mediates the translocation of secretory lysosomes and their fusion with the central portion of the IS, resulting in the release of cytotoxic effector molecules into the immune synaptic cleft [16,18,19,20]. Additionally, Ca^2+^ facilitates the endocytosis of perforin and granzymes and the initiation of apoptotic events in the target cell [18,21,22]. Intracellular Ca^2+^ and kinase signaling downstream of IS formation and IL-2R ligation are thus essential processes for NK cell cytotoxicity. Importantly, phosphatidylinositol-4,5-bisphosphate (PIP_2_) serves as a substrate for phosphoinositide 3-kinase (PI3K) in the IL-2 signaling cascade and phospholipase C (PLC) to liberate Ca^2+^ stores for secretory lysosome translocation and degranulation [23,24,25,26]. Additional crucial proteins that participate in Ca^2+^ signaling events and PIP_2_ interactions include the transient receptor potential (TRP) cation channels.

TRP cation channels are polymodal sensors that detect and mediate cellular responses to physical or chemical perturbations in the extracellular milieu particularly through Ca^2+^ influx [27]. The Ca^2+^- and magnesium (Mg^2+^)-permeable cation channel, TRP melastatin member 7 (TRPM7), is constitutively active and depends on the presence of PIP_2_ to maintain its activity [28,29,30,31]. In turn, TRPM7 modulates the activity of PI3K and PLC according to the availability of intracellular Mg^2+^ [32,33]. TRPM7 additionally possesses an intrinsic kinase domain that modulates the mitogen-activated protein kinase (MAPK) and PI3K-protein kinase B (Akt)-mammalian target of rapamycin (mTOR) pathways common to IL-2R activation, as well as intracellular Ca^2+^ signaling mechanisms that are activated as part of NK cell degranulation events [32,33,34,35]. Hence, the TRPM7 cation channel and kinase domains may influence many key signaling events that involve PIP_2_ and are essential for NK cell activation and effector functions. The role of TRPM7 in NK cell cytotoxicity and interactions with PIP_2_ can be explored pharmacologically using the TRPM7 agonist, naltriben (NTB), and TRPM7 antagonist, NS8593. 

NTB is a specific activator of TRPM7 ionic currents with a reported half-maximal effective concentration (EC_50_) of 18.1 µM to 24 µM for in vitro models characterizing the effect of immediate TRPM7 stimulation [36]. By contrast, NS8593 blocks the function of both the TRPM7 ion channel and kinase domains with a reported half-maximal inhibitory concentration (IC_50_) of 5.9 ± 0.6 µM applied in the presence of 300 µM Mg^2+^ in an in vitro model [37]. A comprehensive review performed by Chubanov and Gudermann summarizes that NS8593 treatment in various cell types replicates the phenotypic features of genetic inactivation of TRPM7 and enables characterization of this channel in human disease [38]. The effect of NS8593 in NK cells, however, was not reported in this review. NTB and NS8593 are therefore appropriate drugs to modulate the activity of TRPM7 to characterize the implications of this protein in the IL-2 priming and cytotoxic mechanisms of NK cells. Importantly, dysregulation of Ca^2+^ signaling and intracellular mechanisms downstream of IL-2R activation that potentially engage TRPM7 may compromise NK cell function. In clinical research, NK cell dysfunction is a consistent finding in several neuroimmune disorders including myalgic encephalomyelitis/chronic fatigue syndrome (ME/CFS).

The pathognomonic features of ME/CFS include fatigue that does not improve with rest, neuroimmune exhaustion, and exacerbation of concomitant symptoms including pain and cognitive impairment after physical or mental exertion [39]. NK cell dysfunction, Ca^2+^ and kinase signaling impairments have been identified in ME/CFS patients [40,41,42,43]. More recently, TRP ion channel aberrations were implicated in the pathomechanism of ME/CFS. The primary findings generated from in vitro models include reduced TRPM3 surface expression and ion channel function, as well as increased TRPM2 surface expression in the NK cells of ME/CFS patients [41,42,44,45,46]. The phylogenetic and biochemical similarities of TRPM7 with TRPM3 and TRPM2 [27], as well as the potential involvement of TRPM7 in cellular signaling mechanisms that are essential to NK cell function, provide a rationale to characterize TRPM7 in the NK cells of ME/CFS patients.

The implication of TRPM7 in NK cell cytotoxicity and the IL-2 priming mechanism in ME/CFS has not been described. Moreover, the efficacy of the IL-2 cytotoxic priming mechanism in the NK cells of ME/CFS was only explored recently, with data suggesting that IL-2 priming also enhances NK cell cytotoxicity in ME/CFS in vitro [47]. This pilot study, therefore, aims to corroborate the effect of IL-2 stimulation in the NK cells of ME/CFS patients. Secondly, this study is the first in vitro investigation to evaluate the potential role of TRPM7 in NK cell cytotoxicity and the co-localization of TRPM7 with PIP_2_ after IL-2 stimulation and pharmacological treatment with NTB or NS8593 in the NK cells of both ME/CFS patients and healthy controls (HCs). The significance of these investigations is to help elucidate the pathomechanism of ME/CFS and identify reliable biomarkers for the diagnosis and treatment of this debilitating disorder.

## 2. Materials and Methods

### 2.1. Study Participants

A total of 17 ME/CFS and 17 HC participants were included in this pilot study. One participant and the matched control were excluded for abnormal full blood count (FBC) results. Participants were identified through the National Center for Neuroimmunology and Emerging Diseases (NCNED) database and ME/CFS support networks between December 2019 and December 2020. Eligible participants were between 18 and 65 years of age and had a body mass index (BMI) between 18.5 kg/m^2^ and 30 kg/m^2^. All ME/CFS patients were classified according to the Canadian Consensus Criteria and the International Consensus Criteria, and age- and sex-matched to healthy controls. Exclusionary criteria were applied to all potential participants and included chronic illness (including autoimmune, cardiac, endocrine, and primary psychiatric disorders), a recent history of smoking, alcohol misuse, pregnancy, or breastfeeding, or administering pharmacological agents that are known to directly or indirectly interfered with TRPM7 channel-kinase activity, Ca^2+^ signaling, or immune cell activity. 

All participants provided written informed consent to participate in the present study approved by the Griffith University Human Research Ethics Committee (HREC/2019/QGC/56469). Eligible participants donated 85 mL of whole venous blood collected in ethylenediaminetetraacetic acid (EDTA) tubes between 7:00 am and 11:00 am in the South-East Queensland, Northern New South Wales, and Melbourne, Victoria regions of Australia. Upon receipt of blood samples to the laboratory, each participant sample was de-identified using a random number and letter generator by an independent blood collector. An FBC was performed within 4 h of collection for each participant as a routine measure for exclusionary hematological abnormalities including features of anemia or white cell derangements.

All participants completed an online questionnaire collecting sociodemographic data, medical history, medications, and symptom history for ME/CFS patients. ME/CFS symptom survey responses were grouped into 10 symptom categories including: (i) cognitive difficulties (slowed thought, impaired concentration, memory consolidation issues); (ii) pain (headaches, muscle pain, multi-joint pain); (iii) sleep disturbances (reversed sleep cycle, disturbed sleep cycle, unrefreshing sleep); (iv) sensory disturbances (sensitivity to touch, vibration, taste, odor, sound; poor coordination or balance); (v) immune disturbances (flu-like symptoms, sore throat, tender lymph nodes); (vi) gastrointestinal disturbances (abdominal pain, nausea, bloating); (vii) cardiovascular disturbances (orthostatic intolerances, lightheadedness, heart palpitations); (viii) respiratory disturbances (difficulty breathing, air hunger); (ix) thermostatic instability (abnormal sweating episodes, hot flushes, cold extremities); and (x) urinary disturbances (changed urination frequency and painful bladder). In addition, a 36-item short-form health survey (SF-36) and World Health Organization (WHO) Disability Assessment Schedule (DAS) were used to determine the level of disability and quality of life.

### 2.2. Peripheral Blood Mononuclear Cell Isolation and Natural Killer Cell Purification

Eighty mL of whole blood was layered over a density gradient medium (Ficoll-Paque Premium; GE Healthcare, Uppsala, Sweden) and peripheral blood mononuclear cells (PBMCs) were isolated by centrifugation as previously described [48]. The total cell count and viability of PBMCs were determined using trypan blue stain (Bio-rad, Hercules, CA, USA) and adjusted to a final concentration of 5 × 10^7^ cells/mL. An EasySep Negative Human NK Cell Isolation Kit (Stemcell Technologies, Vancouver, BC, Canada) was used to separate NK cells from the PBMCs. Isolated NK cells were stained with CD3-APC-H7 (0.5 µg/5 µL) and CD56-Pe-Cy7 (0.25 µg/5 µL) antibodies (BD Bioscience, Miami, FL, USA) for 30 min at room temperature (RT) protected from light and analyzed using the BD Accuri C6 flow cytometer (BD Bioscience, Miami, FL, USA) to assess purity as previously described [47]. Lymphocyte populations were defined using forward scatter and side scatter measures on dot plots generated by the flow cytometer. The NK cell population was defined as CD3^-^ and CD56^+^ as previously described [47]. A purity value of ≥90% for NK cell isolates was deemed acceptable (Appendix A, Figure A1).

### 2.3. Interleukin-2 Stimulation and Drug Treatment

Freshly isolated NK cells were stimulated with 20 IU/mL of recombinant human IL-2 (specific activity 5 × 10^6^ IU/mg) per test (Miltenyi Biotech, BG, Germany). NK cells were additionally treated with NTB (25 µM as determined by dose-response curve, Appendix B, Figure A2) or NTB with NS8593 (10 µM as determined by dose-response curve, Appendix C, Figure A3) (Tocris Bioscience, Sussex, UK) and incubated for 24 h at 37 °C with 5% CO_2_ in Roswell Park Memorial Institute (RPMI)-1640 medium (Invitrogen Life Technologies, Carlsbad, CA, USA) supplemented with 10% fetal bovine serum (FBS) (Invitrogen Life Technologies, Carlsbad, CA, USA). IL-2-stimulated and drug-treated NK cells were washed with RPMI-1640 supplemented with 10% FBS and prepared for NK cell cytotoxic activity (NKCCA) assays or with 1X phosphate-buffered saline (PBS) pH 7.2 (Invitrogen Life Technologies, Carlsbad, CA, USA) for confocal microscopy.

### 2.4. Natural Killer Cell Cytotoxic Activity Assay

NKCCA was assessed at baseline and post-IL-2 stimulation/drug treatment as previously described [47]. NK cells were stained with Paul Karl Horan (PKH)-26 (3.5 µL/test) (Sigma-Aldrich, St. Louis, MO, USA) for 5 min and incubated with human erythromyeloblastoid leukemia (K562) target cells for 4 h at 37 °C with 5% CO_2_ in RPMI-1640 supplemented with 10% FBS. The concentration of NK cells and K562 cells were adjusted to 5 × 10^5^ cells/mL and 1 × 10^6^ cells/mL, respectively, and combined at effector to target ratios (E:T) of 25:1, 12.5:1, and 6.25:1 alongside control conditions for incubation. Control and treated cells were labeled with Annexin V (2.5 µL/test) and 7-AAD (2.5 µL/test) (BD Bioscience, Miami, FL, USA) to assess NKCCA using flow cytometric analysis recording 10,000–30,000 events. NKCCA was calculated as previously described [46] for each E:T as the percent of specific death of K562 cells according to the formula below:$$\mathrm{Cytotoxicity}\;\left(\%\right)=\frac{\left(\mathrm{early}\;\mathrm{stage}\;\mathrm{apoptosis}+\mathrm{late}\;\mathrm{stage}\;\mathrm{apoptosis}+\mathrm{dead}\;K562\;\mathrm{cells}\right)}{\mathrm{All}\;K562\;\mathrm{cells}}\times 100$$

### 2.5. Co-Localization of Transient Receptor Potential Melastatin 7 with Actin and Phosphoinositol 4,5-Bisphosphate in Natural Killer Cells

Confocal microscopy was employed to determine the co-localization of TRPM7 with actin and PIP_2_ post-IL-2 stimulation/drug treatment. All washes and incubations were performed in 1X PBS pH 7.2 at RT and protected from light unless otherwise stated. NK cells were adhered to glass coverslips using Cell-Tak^®^ (BD Bioscience, East Rutherford, NJ, USA) at 2.5 × 10^5^ cells/coverslip. Freshly isolated NK cells were then fixed with 3% paraformaldehyde (Sigma-Aldrich, St Louis, MO, USA) and permeabilized using 0.2% Tween20 (Sigma-Aldrich, St Louis, MO, USA). Blocking was performed using 3% bovine serum albumin (BSA) in 1X PBS for 1 h. NK cells were incubated with primary antibodies against TRPM7 (0.5 µg/mL) (Abcam, Shanghai, China) and PIP_2_ (0.3 µg/mL) (Abcam, Shanghai, China) in 3% BSA overnight at 4 °C. 

Cells were incubated with Alexa Fluor^®^ 488 secondary antibody (1 µg/mL) (Life Technologies, Carlsbad, CA, USA) against the TRPM7 primary and Alexa Fluor^®^ 594 secondary antibody (1 µg/mL) (Life Technologies, Carlsbad, CA, USA) against the PIP_2_ primary for 30 min each. Alexa Fluor^®^ 647 (1:500) (Invitrogen, Carlsbad, CA, USA) was applied for 30 min to stain actin, and 4′,6-diamidino-2-phenylindole (300 nM) (Invitrogen, Carlsbad, CA, USA) was applied for 5 min to stain nuclear material. Coverslips were rinsed in deionized water before mounting on glass slides using Mowiol^®^ 4-88 (Sigma-Aldrich, St Louis, MO, USA) mounting media and allowed to dry at 4 °C overnight. PIP_2_ immunofluorescence was confirmed in non-permeabilized NK cells and TRPM7 immunofluorescence was confirmed using human TRPM7 blocking peptide (Abcam, Shanghai, China). Images were acquired using a Nikon A1 laser scanning confocal microscope (Nikon, Tokyo, Japan) under 60× magnification (Appendix D and Appendix E, Figure A4 and Figure A5). Pearson’s and Mander’s correlation coefficients as well as K1 and K2 values were obtained for approximately 8 cells per condition per patient using NIS-Elements image analysis software (Nikon, Tokyo, Japan).

### 2.6. Statistical Analysis

Pilot data from this investigation were analyzed using GraphPad Prism version 8.0 (GraphPad Software Inc., Version 8, La Jolla, CA, USA). Data distribution and normality were assessed using the Shapiro-Wilk normality test, skewness, and kurtosis. Statistical significance between and within groups for NKCCA and confocal co-localization was assessed using the independent Mann-Whitney U test. Significance was set at *p* < 0.05 and all data are presented as the mean ± standard error of the mean (SEM) unless otherwise stated.

## 3. Results

### 3.1. Participant Demographics

A total of 34 participants (17 HCs and 17 ME/CFS patients) were included in the present study. No significant differences in age, gender, and BMI were identified between groups (Table 1). ME/CFS patients reported greater impairment in all SF-36 items and the WHODAS items, except participation in school/work, compared with HCs. All included participants were within the reference ranges for each FBC parameter and there were no significant differences between groups for these parameters.

### 3.2. Baseline Natural Killer Cell Cytotoxic Activity

The baseline NKCCA at the E:T of 25:1 for HCs was 28.94 ± 3.893 and 17.13 ± 3.721 for ME/CFS patients, respectively (*p* < 0.05); at 12.5:1 for HCs was 11.65 ± 1.821 and 7.19 ± 2.120 for ME/CFS patients, respectively (*p* < 0.05); and at 6.25:1 for HCs was 7.0 ± 1.260 and 2.524 ± 1.182 for ME/CFS patients, respectively (*p* < 0.001) (Figure 1). At baseline, NK cell cytotoxicity was significantly reduced in the ME/CFS cohort compared with the HC cohort for all E:Ts (25:1, 12.5:1, 6.25:1).

### 3.3. Natural Killer Cell Cytotoxic Activity Post-Interleukin-2 Stimulation and Drug Treatment

The NKCCA was significantly increased after IL-2 (20 IU/mL) stimulation with the E:T of 25:1 for HCs being 49.67 ± 6.273 and 48.55 ± 5.129 for ME/CFS patients, respectively (*p* > 0.05); at 12.5:1 for HCs being 49.42 ± 7.256 and 52.68 ± 5.998 for ME/CFS patients, respectively (*p* > 0.05); and at 6.25:1 for HCs being 39.15 ± 6.347 and 39.08 ± 5.034 for ME/CFS patients, respectively (*p* > 0.05). Significant differences between baseline lysis and the IL-2 stimulated lysis results within groups were identified (Figure 2). No significant difference was identified between the IL-2 control and the TRPM7 agonism (20IU/mL IL-2 + 25 µM NTB) or TRPM7 antagonism (20 IU/mL IL-2 + 25 µM NTB + 10 µM NS8593) conditions between or within groups for any of the different E:Ts (Figure 3). 

### 3.4. Co-Localization of Transient Receptor Potential Melastatin 7 with Actin

No significant differences were observed for the co-localization of TRPM7 with actin following IL-2 (20 IU/mL) stimulation and treatment with the TRPM7 agonist (20 IU/mL IL-2 + 25 µM NTB) within or between groups. A significant difference between the TRPM7 agonist (20 IU/mL IL-2 + 25 µM NTB) and TRPM7 antagonist (20 IU/mL IL-2 + 25 µM NTB + 10 µM NS8593) treatments was observed within the ME/CFS cohort (*p* < 0.001) specifically showing an increase in TRPM7 co-localization with actin with TRPM7 antagonism. Additionally, the ME/CFS cohort showed a significant increase in TRPM7 co-localization with actin for the TRPM7 antagonism treatment compared with HCs (*p* < 0.01) (Figure 4). 

### 3.5. Co-Localization of Transient Receptor Potential Melastatin 7 with Phosphatidylinositol 4,5-Bisphosphate

No significant differences were observed for the co-localization of TRPM7 with PIP_2_ following IL-2 (20 IU/mL) stimulation and treatment with the TRPM7 agonist (20 IU/mL IL-2 + 25 µM NTB) within or between groups. A significant difference between the TRPM7 agonist (20 IU/mL IL-2 + 25 µM NTB) and TRPM7 antagonist (20 IU/mL IL-2 + 25 µM NTB + 10 µM NS8593) treatments was observed within the HC cohort (*p* < 0.01) and ME/CFS cohort (*p* < 0.05) specifically showing an increase in TRPM7 co-localization with PIP_2_ with TRPM7 antagonism (Figure 5).

## 4. Discussion

This pilot study supports previous investigations demonstrating significant reductions in baseline NKCCA for ME/CFS patients compared with HCs [49,50,51,52,53,54]. Moreover, this study confirms a previous investigation of significantly enhanced NKCCA following overnight IL-2 stimulation compared with baseline NKCCA in both ME/CFS patients and HCs [47]. This study is novel as it reports a significantly higher co-localization of TRPM7 with actin in the ME/CFS cohort following NS8593 treatment compared with the NTB condition for ME/CFS and the NS8593 condition for HC. For the first time, this study also reports a significant increase in co-localization of TRPM7 with PIP_2_ in the HC and ME/CFS cohorts following NS8593 treatment compared with the HC NTB and ME/CFS NTB conditions, respectively. The present investigation proposes that crosstalk between intracellular signaling pathways common to IL-2 and TRPM7 may exist.

Importantly, overnight IL-2 incubation of NK cells significantly increased NKCCA in both ME/CFS patients and HCs. This suggests that dysregulation of the IL-2 signaling pathways may not explain the significant reduction in NKCCA that is a consistent feature in ME/CFS patients [49]. IL-2R activation promotes dimerization of the cytoplasmic receptor domains and activation of Janus kinase 1 (JAK1) and JAK3. Signal transducer and activator of transcription 5 (STAT5) is subsequently phosphorylated, which promotes dimerization and translocation of STAT5 to the nucleus. JAK1 additionally activates the MAPK pathway and JAK3 activates the PI3K-Akt-mTOR pathway. 

This intracellular signaling cascade is common to TRPM7 kinase signaling and therefore suggests that signaling pathways downstream of IL-2R ligation are shared by TRPM7 and IL-2 [55,56]. Previous investigations have reported impaired Ca^2+^ storage [41] and intracellular signaling [43] in the NK cell function of ME/CFS patients compared with HCs. The findings of this study suggest that Ca^2+^ entry through TRPM7 and IL-2 binding to its cognate receptor may provide a crosstalk-dependent pathway for NK cell function. Recently, Eaton-Fitch et al. [47] reported TRPM3 and PIP_2_ dependent pathway interactions in ME/CFS patients, with a corresponding increase in NKCCA in both ME/CFS and HC following overnight incubation with IL-2.

This study is the first to report that TRPM7 is expressed by NK cells using immunofluorescence and confocal microscopy techniques. Additionally, this study reports the novel finding of TRPM7 co-localizing with actin and PIP_2_ following NS8593 treatment in ME/CFS patients and HCs. This may likely be due to inhibition of the TRPM7 ion channel and kinase domains necessitating increased co-localization of TRPM7 with PIP_2_, possibly in an attempt by the cell to maintain TRPM7 activity. Activated PLC hydrolyzes and thereby depletes PIP_2_ by catalyzing the formation of inositol triphosphate and diacylglycerol, which promote intracellular Ca^2+^ store release and MAPK signaling cascades, respectively [57]. PLC and MAPK signaling networks regulate cellular mechanisms that recruit and activate other signaling proteins, reorganize the cytoskeleton, mobilize Ca^2+^, and activate NK cell effector functions [58,59,60,61]. 

It is known that the availability of PIP_2_ influences the activity of TRPM7 according to a biphasic model. Transient depletion of PIP_2_ from the activation of PLC potentiates TRPM7 ionic currents, and exhaustion of PIP_2_ reserves beyond an unknown limit renders TRPM7 inactive [62,63,64,65,66,67,68]. In turn, TRPM7 modulates PLC activity and Ca^2+^ signaling. Under conditions of low intracellular Mg^2+^ levels, the TRPM7 kinase phosphorylates PLC, thereby downregulating Ca^2+^ signaling [32]. Hence, significantly increased co-localization of TRPM7 with PIP_2_ following inhibition of the channel and kinase domains may indicate a mechanism to re-establish TRPM7 activity, which is speculated to be more effective in HCs compared with ME/CFS patients due to dysregulation of Ca^2+^-dependent processes including protein trafficking.

From this investigation, it was also found that TRPM7 co-localization potentially increases to support cell surface signaling mechanisms such as Ca^2+^ influx, exocytosis, and kinase signaling. Alternatively, this finding may be explained as an artifact from TRPM7 co-localizing with PIP_2_, which is more abundant at the plasma membrane. Interestingly, Balinas et al., reported a significant increase in TRMP2 as a possible compensatory mechanism for reduced NKCCA in ME/CFS patients compared to HCs [46]; however, PIP_2_ was not assessed. 

This current investigation additionally reported significant changes in co-localization of TRPM7 with cortical actin of NK cells in HCs, as well as ME/CFS patients, to a lesser extent. TRPM7 expression on the surface of cell membranes is normally scarce [69] although comparatively greater in cells derived from the hematopoietic lineage, including lymphocytes, especially when activated [70]. Rather, TRPM7 predominantly exists on the surface of intracellular vesicles [69] and may be mobilized to the surface when this channel needs to be replenished. 

Classically, the soluble N-ethylmaleimide-sensitive-factor accessory protein-receptor (SNARE) proteins are key mediators that facilitate vesicle packaging and fusion at the cell membrane and require adequate intracellular Ca^2+^ concentrations to function [71]. Thus, translocation of TRP ion channels, in this instance TRPM7, may be supported by local increases in Ca^2+^. Differences in co-localization of TRPM7 in ME/CFS compared with HCs may therefore reflect the known Ca^2+^ dysregulation in ME/CFS and supported by previous dysfunctional TRPM3 and Ca^2+^ signaling findings reported in ME/CFS patients [41,42,44,45,46,47,72]. Future investigations are indicated to determine whether compromised Ca^2+^ mobilization in ME/CFS patients is indeed responsible for impaired protein translocation and expression.

Limitations of this pilot study include the finite number of cells that were able to be isolated from donor blood samples for experimental work. Consequentially, this restricted the scope of outcomes that could be addressed by the present study. Further investigations examining the intracellular signaling pathways associated with TRPM7 may elucidate the findings reported in this current project. Moreover, no pre-existing literature regarding the effect of treatment with NS8593 alone in NK cells was available. While potential off-target effects may exist for NS8593 and NTB, there is a paucity of literature in NK cells. Using the latest literature, NS8593 and NTB were deemed the most appropriate pharmacological tools for characterizing TRPM7 channel activity in the NK cells of ME/CFS patients. Extending this pilot investigation with further research to characterize TRPM7 activity in the NK cells of ME/CFS patients may, therefore, augment the current hypothesis of pathophysiological mechanisms in ME/CFS.

## 5. Conclusions

The results from this investigation are preliminary and suggest crosstalk between IL-2 and TRMP7 exists. However, a larger sample size to confirm these findings is warranted. Importantly, these current data, along with previous studies reporting TRP dysfunction and altered expression patterns in the NK cells of ME/CFS patients provide evidence for potential channelopathy as the pathomechanism for ME/CFS. Prospective Ca^2+^ influx assays, isotyping, and patch-clamp electrophysiology provide additional avenues for further characterizing TRPM7 in the pathomechanism of this illness for possible targeted pharmacotherapeutic interventions for the treatment of ME/CFS.

## Figures and Tables

**Figure 1 ijerph-18-11879-f001:**
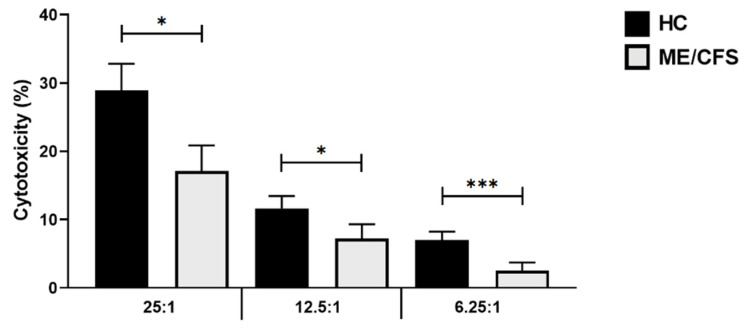
Baseline NK cell cytotoxicity; *n* = 17 for HC and *n* = 17 for ME/CFS. A significantly lower NKCCA was observed in ME/CFS patients compared with HCs at baseline for each effector to target ratio. Data are represented as the mean ± SEM as determined by independent Mann-Whitney U tests. * *p* < 0.05, *** *p* < 0.001. Abbreviations: healthy control (HC), myelogenous leukemia target cell line (K562), myalgic encephalomyelitis/chronic fatigue syndrome (ME/CFS), natural killer (NK), natural killer cell cytotoxic activity (NKCAA).

**Figure 2 ijerph-18-11879-f002:**
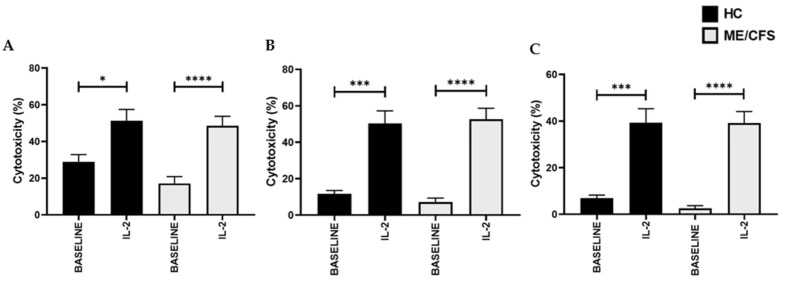
NK cell cytotoxicity at baseline compared with post-IL-2 stimulation; *n* = 17 for HC and *n* = 17 for ME/CFS. (**A**) 25:1 ratio (**B**) 12.5:1 ratio (**C**) 12.5:1 ratio. A significantly higher NKCCA was observed in both HCs and ME/CFS patients following IL-2 (20 IU/mL) stimulation compared with that at baseline for each effector to target ratio. Data are represented as the mean ± SEM as determined by independent Mann-Whitney U tests. * *p* < 0.05, *** *p* < 0.001, **** *p* < 0.0001. Abbreviations: healthy controls (HCs), interleukin-2 (IL-2), myelogenous leukemia target cell line (K562), myalgic encephalomyelitis/chronic fatigue syndrome (ME/CFS), natural killer (NK), natural killer cell cytotoxic activity (NKCAA).

**Figure 3 ijerph-18-11879-f003:**
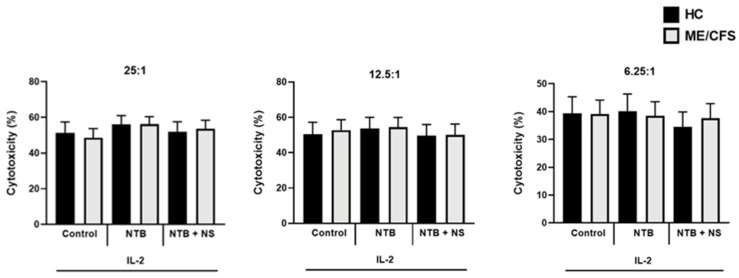
NK cell cytotoxicity post-IL-2 stimulation compared with TRPM7 drug treatments in HC vs. ME/CFS; *n* = 17 for HC and *n* = 17 for ME/CFS. After IL-2 (20 IU/mL) stimulation and TRPM7 agonist (20 IU/mL IL-2 + 25 µM NTB) or antagonist (20 IU/mL IL-2 + 25 µM NTB + 10 µM NS8593) treatments for 24 h, no significant differences were observed within or between groups for each condition for any effector to target ratio. Data are represented as the mean ± SEM as determined by independent Mann-Whitney U tests. Abbreviations: healthy controls (HCs), interleukin-2 (IL-2), myelogenous leukemia target cell line (K562), myalgic encephalomyelitis/chronic fatigue syndrome (ME/CFS), natural killer (NK), NS8593 (NS), naltriben (NTB).

**Figure 4 ijerph-18-11879-f004:**
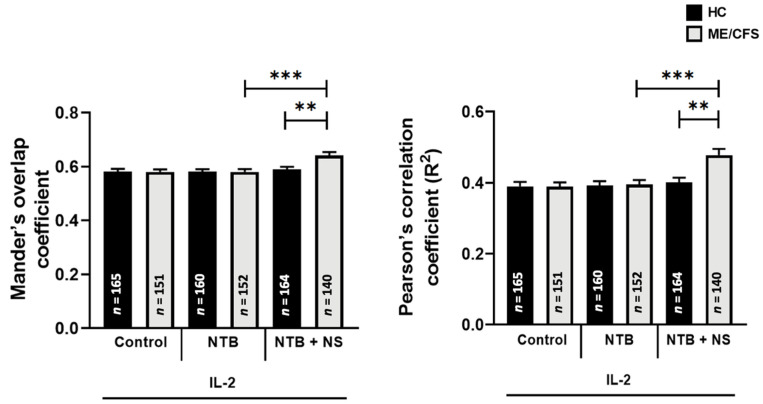
Co-localization of TRPM7 with actin in NK cells following IL-2 stimulation and treatment with NTB and NS8593 in HC vs. ME/CFS; *n* = 17 for HC and *n* = 17 for ME/CFS. The number of cells analyzed per condition are depicted on the bar graphs: *n* = 165 for HC Control, *n* = 151 for ME/CFS control, *n* = 160 for HC NTB, *n* = 152 for ME/CFS NTB, *n* = 164 for HC NTB + NS, *n* = 140 for ME/CFS NTB + NS. A significantly higher co-localization of TRPM7 with actin was identified in the ME/CFS cohort following NS8593 treatment compared with the ME/CFS NTB and HC NS8593 conditions. This difference was not present in the HC cohort. Data are represented as the mean ± SEM as determined by independent Mann-Whitney U tests. ** *p* < 0.01, *** *p* < 0.001. Abbreviations: healthy control (HC), myalgic encephalomyelitis/chronic fatigue syndrome (ME/CFS), natural killer (NK), NS8593 (NS), naltriben (NTB), transient receptor potential melastatin 7 (TRPM7).

**Figure 5 ijerph-18-11879-f005:**
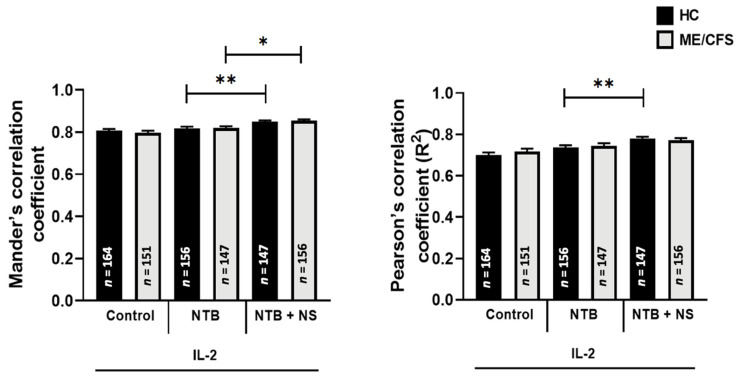
Co-localization of TRPM7 with PIP_2_ in NK cells following IL-2 stimulation and treatment with NTB and NS8593 in HC vs. ME/CFS; *n* = 17 for HC and *n* = 17 for ME/CFS. The number of cells analyzed per condition are depicted on the bar graphs: *n* = 165 for HC Control, *n* = 151 for ME/CFS Control, *n* = 160 for HC NTB, *n* = 152 for ME/CFS NTB, *n* = 164 for HC NTB + NS, *n* = 140 for ME/CFS NTB + NS. A significantly higher co-localization of TRPM7 with PIP_2_ was identified in the HC and ME/CFS cohorts following NS8593 treatment compared with the HC NTB and ME/CFS NTB conditions, respectively. No significant differences were identified between groups for each condition. Data are presented as the mean ± SEM as determined by independent Mann-Whitney U tests. * *p* < 0.05, ** *p* < 0.01. Abbreviations: healthy control (HC), myalgic encephalomyelitis/chronic fatigue syndrome (ME/CFS), natural killer (NK), NS8593 (NS), phosphoinositide 4,5-bisphosphate (PIP_2_), transient receptor potential melastatin 7 (TRPM7).

**Table 1 ijerph-18-11879-t001:** Participant demographic and full blood count results; *n* = 17 for HC and *n* = 17 for ME/CFS. Data are represented as mean ± SEM using Mann-Whitney U tests. *** *p* < 0.001, **** *p* < 0.0001. Abbreviations: healthy control (HC), myalgic encephalomyelitis (ME/CFS), body mass index (BMI), World Health Organization disability assessment schedule (WHODAS), short-form health survey (SF-36).

Category	Item	HC	ME/CFS	*p*-Value
General demographics	Age (years)	48.1 ± 2.1	48.5 ± 2.0	0.7264
	Gender			
	Male (%, n)	23.5, 5	23.5, 5	
	Female (%, n)	76.5, 13	76.5, 13	
	BMI (kg/m^2^)	23.9 ± 0.9	25.2 ± 1.0	0.3435
WHODAS	Understanding and communication	7.1 ± 2.0	41.6 ± 3.1	****
	Mobility	2.4 ± 1.4	38.2 ± 6.2	****
	Self-care	0.4 ± 0.4	18.4 ± 4.9	***
	Interpersonal relationships	3.3 ± 1.7	33.5 ± 5.0	****
	Life activities	7.4 ± 3.5	62.5 ± 6.2	****
	Participation in work/school	7.4 ± 3.7	12.6 ± 5.1	0.4555
	Participation in society	4.4 ± 1.9	52.2 ± 5.1	****
Illness demographic (SF-36)	Pain (%)	90.7 ± 3.2	37.5 ± 5.1	****
	Physical functioning (%)	92.7 ± 4.7	47.6 ± 6.4	****
	Role physical (%)	93.8 ± 2.7	25.0 ± 5.5	****
	General health (%)	80.6 ± 2.7	27.2 ± 3.3	****
	Social functioning (%)	89.7 ± 6.2	27.2 ± 4.7	****
	Role emotional (%)	95.1 ± 2.4	63.7 ± 6.9	***
	Wellbeing	74.8 ± 3.5	43.9 ± 2.8	****
Full blood count	White blood cells	6.2 ± 0.5	5.7 ± 0.3	0.5454
	Lymphocytes	1.70 ± 0.15	1.85 ± 0.09	0.1243
	Neutrophils	3.77 ± 0.39	3.26 ± 0.27	0.3524
	Monocytes	0.49 ± 0.05	0.42 ± 0.02	0.2778
	Eosinophils	0.13 ± 0.02	0.14 ± 0.02	0.8048
	Basophils	0.04 ± 0.00	0.04 ± 0.00	0.3378
	Platelets	265 ± 19	242 ± 9	0.2085
	Red blood cells	4.51 ± 0.09	4.57 ± 0.12	0.6645
	Hematocrit	0.40 ± 0.01	0.41 ± 0.01	0.3416
	Hemoglobin	131 ± 3	137 ± 3	0.2628

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
