# Peer review of "Characterization of IL-2 Stimulation and TRPM7 Pharmacomodulation in NK Cell Cytotoxicity and Channel Co-Localization with PIP2 in Myalgic Encephalomyelitis/Chronic Fatigue Syndrome Patients"

_ijerph, 2021, doi:10.3390/ijerph182211879_

Round 1
Reviewer 1 Report
I have concerns about the activity of NS8593 alone on NK cells. I apologize if I have missed this in your paper.
Figure 1 shows that baseline unstimulated NK cell cytotoxicity is significantly higher in controls than ME/CFS.
What are the effect sizes? This is a challenge to predict from SEM without knowing n. The relevance is to ensure that future studies are powered appropriately for future studies.
Do NTB and NS8593 have any effects at baseline (no IL2 incubation)?
Do these drugs increase cytotoxicity in ME/CFS NK cells?
Does NS8593 alone have any effect on its own at baseline or after IL2 incubation in ME/CFS or control subjects? This critical control
is needed for all experiments. Have those experiments been performed before and reported elsewhere?
Figure 2 shows that IL2 incubation (priming) increased NK cytotoxicity in control and ME/CFS.
Figure 3 shows that the TRPM7 agonist NTB has no effect alone, and that the addition of NS8593 to NTB had no effect. It is not clear if NS8593 alone could have an effect that is antagonized by NTB.
What is the figure under Section 3.4? There is no legend.
Figure 4 shows the co-expression of TRPM7 and actin after IL2. The addition of NS8593&NTB increases the overlap of TRPM7 and actin in ME/CFS and control. Is this an effect of NS8593 alone that is not related to the (absent) effect of NTB? The response to NS8593&NTB NS8593&NTB appears to be greater than the (presumably no drug) control in ME/CFS.
Figure 5 shows colocalization of TRPM7 and PIP2 after IL2. NS8593&NTB increase cytotoxicity compared to NTB alone (and possible drug free control?) based on Marker’s overlap in both control and ME/CFS. NTB and NS8593&NTB had no effect on cytotoxicity after IL2 incubation in MECFS or control. The effects at baseline were not reported here. NS8593 appears to increase co-localization of TRPM7 with actin in ME/CFS but not control, and TRPM7 and PIP2 in both ME/CFS and control. NTB appears to have no effects. However, there are no studies of NS8593 alone to investigate off target activity on K(Ca)2 SK channels or other potential targets. This is intriguing because NS8593 may modulate divalent cation flux in the
intestine, including Zn2+, Mg2+, Ca2+, and potentially Mn2+. If NS8593 has effects of its own, then this may open up new hypotheses of ME/CFS pathological mechanisms.
PubMed Mittermeier L, Demirkhanyan L,
Stadlbauer B, Breit A, Recordati C, Hilgendorff A, Matsushita M, Braun A, Simmons DG, Zakharian E, Gudermann T, Chubanov V. TRPM7 is the central gatekeeper of intestinal mineral absorption essential for postnatal survival. Proc Natl Acad Sci U S A. 2019 Mar 5;116(10):4706-4715. doi: 10.1073/pnas.1810633116. Epub 2019 Feb 15. PMID: 30770447; PMCID: PMC6410795.
Abstract
Zn2+, Mg2+, and Ca2+ are essential minerals required for a plethora of metabolic processes and signaling pathways. Different categories of cationselective channels and transporters are therefore required to tightly control the cellular levels of individual metals in a cell-specific manner. However, the mechanisms responsible for the organismal balance of these essential minerals are poorly understood. Herein, we identify a central and indispensable role of the channel-kinase TRPM7 for organismal mineral homeostasis. The function of TRPM7 was assessed by single-channel analysis of TRPM7, phenotyping of TRPM7-deficient cells in conjunction with metabolic profiling of mice carrying kidney-and intestine-restricted null mutations in Trpm7 and animals with a global "kinase-dead" point mutation in thegene. The TRPM7 channel reconstituted in lipid bilayers displayed a similar permeability to Zn2+ and Mg2+Consistently, we found that endogenous TRPM7 regulates the total content of Zn2+ and Mg2+ in cultured cells.Unexpectedly, genetic inactivation of intestinal rather than kidney TRPM7 caused profound deficienciesspecifically of Zn2+, Mg2+, and Ca2+ at the organismal level, a scenario incompatible with early postnatal growthand survival. In contrast, global ablation of TRPM7 kinase activity did not affect mineral homeostasis, reinforcingthe importance of the channel activity of TRPM7. Finally, dietary Zn2+ and Mg2+ fortifications significantly extended the survival of offspring lacking intestinal TRPM7. Hence, the organismal balance of divalent cationscritically relies on one common gatekeeper, the intestinal TRPM7 channel.
Chubanov V, Mederos y Schnitzler M,Meißner M, Schäfer S, Abstiens K, Hofmann T, Gudermann T. Natural and synthetic modulators of SK (K(ca)2)potassium channels inhibit magnesium-dependent activity of the kinase-coupled cation channel TRPM7. Br JPharmacol. 2012 Jun;166(4):1357-76. doi: 10.1111/j.1476-5381.2012.01855.x. PMID: 22242975; PMCID:PMC3417452.
Abstract
Background and purpose: Transient receptor potential cation channel subfamily Mmember 7 (TRPM7) is a bifunctional protein comprising a TRP ion channel segment linked to an α-type proteinkinase domain. TRPM7 is essential for proliferation and cell growth. Up-regulation of TRPM7 function is involvedin anoxic neuronal death, cardiac fibrosis and tumour cell proliferation. The goal of this work was to identify non-toxic inhibitors of the TRPM7 channel and to assess the effect of blocking endogenous TRPM7 currents on thephenotype of living cells.
Experimental approach: We developed an aequorin bioluminescence-based assay ofTRPM7 channel activity and performed a hypothesis-driven screen for inhibitors of the channel. The candidatesidentified were further assessed electrophysiologically and in cell biological experiments.
Key results: TRPM7currents were inhibited by modulators of small conductance Ca²⁺ -activated K⁺ channels (K(Ca)2.1-2.3; SK)channels, including the antimalarial plant alkaloid quinine, CyPPA, dequalinium, NS8593, SKA31 and UCL 1684.The most potent compound NS8593 (IC₅₀ 1.6 μM) specifically targeted TRPM7 as compared with other TRPchannels, interfered with Mg²⁺ -dependent regulation of TRPM7 channel and inhibited the motility of culturedcells. NS8593 exhibited full and reversible block of native TRPM7-like currents in HEK 293 cells, freshly isolatedsmooth muscle cells, primary podocytes and ventricular myocytes.
Conclusions and implications: This studyreveals a tight overlap in the pharmacological profiles of TRPM7 and K(Ca)2.1-2.3 channels. NS8593 acts as anegative gating modulator of TRPM7 and is well-suited to study functional features and cellular roles ofendogenous TRPM7.
Pedarzani P, Stocker M. Molecular and cellular basis of small--and intermediate-conductance, calcium-activated potassium channel function in the brain. Cell Mol Life Sci. 2008 Oct;65(20):3196-217. doi: 10.1007/s00018-008-8216-x. PMID: 18597044; PMCID: PMC2798969.
Abstract
Small conductancecalcium-activated potassium (SK or K(Ca)2) channels link intracellular calcium transients to membrane potentialchanges. SK channel subtypes present different pharmacology and distribution in the nervous system. Theselective blocker apamin, SK enhancers and mice lacking specific SK channel subunits have revealedmultifaceted functions of these channels in neurons, glia and cerebral blood vessels. SK channels regulateneuronal firing by contributing to the afterhyperpolarization following action potentials and mediating I(AHP), andpartake in a calcium-mediated feedback loop with NMDA receptors, controlling the threshold for induction ofhippocampal long-term potentiation. The function of distinct SK channel subtypes in different neurons oftenresults from their specific coupling to different calcium sources. The prominent role of SK channels in themodulation of excitability and synaptic function of limbic, dopaminergic and cerebellar neurons hints at theirpossible involvement in neuronal dysfunction, either as part of the causal mechanism or as potential therapeutictargets.
Strøbaek D, Hougaard C, Johansen TH, Sørensen US, Nielsen EØ, Nielsen KS, Taylor RD, Pedarzani P,Christophersen P. Inhibitory gating modulation of small conductance Ca2+-activated K+ channels by thesynthetic compound (R)-N-(benzimidazol-2-yl)-1,2,3,4-tetrahydro-1-naphtylamine (NS8593) reducesafterhyperpolarizing current in hippocampal CA1 neurons. Mol Pharmacol. 2006 Nov;70(5):1771-82. doi:10.1124/mol.106.027110. Epub 2006 Aug 22. PMID: 16926279.
Abstract
SK channels are small conductanceCa(2+)-activated K(+) channels important for the control of neuronal excitability, the fine tuning of firing patterns,and the regulation of synaptic mechanisms. The classic SK channel pharmacology has largely focused on the
peptide apamin, which acts extracellularly by a pore-blocking mechanism. 1-Ethyl-2-benzimidazolinone (1-EBIO)and 6,7-dichloro-1H-indole-2,3-dione 3-oxime (NS309) have been identified as positive gating modulators thatincrease the apparent Ca(2+) sensitivity of SK channels. In the present study, we describe inhibitory gatingmodulation as a novel principle for selective inhibition of SK channels. In whole-cell patch-clamp experiments,the compound (R)-N-(benzimidazol-2-yl)-1,2,3,4-tetrahydro-1-naphtylamine (NS8593) reversibly inhibitedrecombinant SK3-mediated currents (human SK3 and rat SK3) with potencies around 100 nM. However, incontrast to known pore blockers, NS8593 did not inhibit (125)I-apamin binding. Using excised patches, it wasdemonstrated that NS8593 decreased the Ca(2+) sensitivity by shifting the activation curve for Ca(2+) to theright, only slightly affecting the maximal Ca(2+)-activated SK current. NS8593 inhibited all the SK1-3 subtypesCa(2+)-dependently (K(d) = 0.42, 0.60, and 0.73 microM, respectively, at 0.5 microM Ca(2+)), whereas thecompound did not affect the Ca(2+)-activated K(+) channels of intermediate and large conductance (hIK andhBK channels, respectively). The site of action was accessible from both sides of the membrane, and theNS8593-mediated inhibition was prevented in the presence of a high concentration of the positive modulatorNS309. NS8593 was further tested on mouse CA1 neurons in hippocampal slices and shown to inhibit theapamin and tubocurarine-sensitive SK-mediated afterhyperpolarizing current, at a concentration of 3 microM.
Reviewer 2 Report
The paper identifies a novel mechanism of NK cell dysfunction in ME/CFS. The main aim as stated by the authors is to warrant an in-depth study to further explore the role of TRPM7 in NK dysfunction in ME/CFS. While is paper has very promising preliminary data there are few concerns:
Major:
- In the results section Figure 1: Comparison between HC and ME/CFS is not shown. As mentioned in the introduction increased cytotoxicity post-IL-2 was expected. If the authors intend to depict that this is the case in ME/CFS as well then stating it would greatly benefit the reader.
- Figure 3: A and B are essentially used to generate C hence only depicting C would be sufficient.
- Labels in Figure 4 are not correct. A is repeated twice and the figures appear to be repeated. Further similar to Figure 3, only C will be sufficient.
- One of the major concerns is the lack of images for co-localization data. Images for HC as well as ME/CSF would make the data stronger.
- The discussion mentions multiple signaling pathways that can be altered instead of IL-2 signal mediated NKCCA, however, the authors themselves do not confirm these. This is the same with the downstream effects of TRPM7. While the current data is speculated to depict alterations in major signaling cascades the authors have not confirmed any of them. The addition of this data will make the study stronger and argue the need for further investigation.
Overall, this paper offers a potential therapeutic research direction for ME/CFS patients. Further investigation is necessary to validate and confirm these findings, but this paper depicts the importance of future research on NK cells and TRPM7 in ME/CFS.
Round 2
Reviewer 2 Report
The authors have addressed all the concerns and made the necessary changes. The manuscript can be accepted in its current form. Congratulations. Hope this paper does promote more research in this field.